# Craniofacial Phenotype in Obstructive Sleep Apnea and Its Impact on Positive Airway Pressure (PAP) Adherence

**DOI:** 10.3390/jpm13081196

**Published:** 2023-07-27

**Authors:** Jae-Seon Park, Bin Kwon, Hyun-Seok Kang, Seong-Jin Yun, Sung-Jun Han, Yeso Choi, Sung-Hun Kang, Mi-Yeon Lee, Kyung-Chul Lee, Seok-Jin Hong

**Affiliations:** 1Department of Otorhinolaryngology-Head and Neck Surgery, Kangbuk Samsung Hospital, Sungkyunkwan University School of Medicine, Seoul 03181, Republic of Korea; blackfaced1@gmail.com (J.-S.P.); hyunsok.kang@samsung.com (H.-S.K.); seongjin1.yun@samsung.com (S.-J.Y.); malice23@nate.com (S.-H.K.); kyungchul.lee@samsung.com (K.-C.L.); 2Department of Otorhinolaryngology-Head and Neck Surgery, Dongtan Sacred Heart Hospital, Hallym University College of Medicine, Hwaseong 18450, Republic of Korea; binkwon124@naver.com (B.K.); gtw0917@nate.com (S.-J.H.); 201080@hallym.or.kr (Y.C.); 3Division of Biostatistics, Department of R&D Management, Kangbuk Samsung Hospital, Sungkyunkwan University School of Medicine, Seoul 03181, Republic of Korea; my7713.lee@samsung.com

**Keywords:** sleep apnea, obstructive, continuous positive airway pressure, treatment adherence and compliance, polysomnography, tomography, X-ray computed, cephalometry, hyoid bone

## Abstract

Positive airway pressure (PAP) is an important treatment tool for patients with moderate and severe obstructive sleep apnea (OSA), and adherence to PAP significantly affects treatment outcomes. Disease severity, adverse effects, and psychosocial factors are known to predict medication adherence. Cephalometric parameters have been reported to positively correlate with upper airway collapse. However, research on the correlation between these cephalometric parameters and PAP adherence remains insufficient. This study aimed to identify this relationship. This study included 185 patients with OSA who were prescribed PAP. Polysomnography (PSG) was performed to diagnose OSA, and paranasal sinus computed tomography (PNS CT) was performed to check for comorbidities of the upper airway. In addition, cephalometric parameters such as the hyoid–posterior nasal spine (H-PNS), posterior nasal spine–mandibular plane (PNS-MP), and hyoid–mandibular plane (H-MP) were measured in the midsagittal and axial CT views. Adherence was evaluated 3–12 months after the PAP prescription. A total of 136 patients were PAP-adherent, and 49 were nonadherent. There were more males in the adherent group and a higher average height in the adherent group. The PSG results showed that the apnea–hypopnea index (AHI), respiratory disturbance index (RDI), oxygen desaturation index (ODI), arousal index (AI), rapid eye movement (REM) AHI, and supine AHI were significantly higher, and the lowest oxygen saturation was lower in the adherent group. In the analysis of covariance (ANCOVA) model adjusted for sex and height, among the cephalometric parameters, H-MP was significantly longer in the adherent group (*p* = 0.027), and H-PNS showed a longer tendency (*p* = 0.074). In the logistic regression analysis model, the odds ratio (OR) and 95% confidence intervals (95% CI) of adherence and severe OSA in the third tertile compared to the first tertile of H-MP were 2.93 (1.25–6.86) and 4.00 (1.87–8.56). In the case of H-PNS, they were 2.58 (1.14–5.81) and 4.86 (2.24–10.54), respectively. This study concluded that an inferiorly placed hyoid bone in adult patients is associated with PAP adherence and disease severity.

## 1. Introduction

Obstructive sleep apnea (OSA) is a common disease with a prevalence of 2–4% in the adult population [1] and is associated with excessive daytime sleepiness, reduced quality of life, an increased risk of traffic accidents, and cardiovascular disease [2].

Positive airway pressure (PAP) is recommended as the standard treatment for moderate-to-severe OSA and can be performed as an option for mild OSA [1]. Recent guidelines recommend PAP in cases of excessive sleepiness, decreased sleep-related quality of life, or hypertension [3]. However, despite the good therapeutic effect of PAP, adherence is not high and has been reported to be between 40% and 85% [4]. In several observational studies, PAP adherence was defined when devices were used for ≥4 h/day on at least 70% of days, and this was considered appropriate treatment [5,6,7]. Factors influencing PAP adherence include sociodemographic characteristics, disease severity, psychosocial factors, and adverse effects [4].

Cephalometric parameters are commonly used to analyze the anatomical characteristics of the upper respiratory tract in patients with OSA. Cephalometric evaluation using computed tomography (CT) can provide information on the measurement of the upper airway, position of the hyoid bone, shape of the mandible, posterior airway space, shape of the tongue, and thickness of the palate and can be used to predict the correlation between sleep apnea and the upper airway [8,9]. Among them, the hyoid-mandibular plane (H-MP) and hyoid–posterior nasal spine (H-PNS), which are indicators related to the location of the hyoid, and soft palate length (SPL) show a significant correlation with pharyngeal critical pressure, which is an indicator of upper airway collapsibility [8]. In addition, it has been reported that there was a difference between OSA patients and normal subjects when measuring parameters related to the contour of the retroglossal airway in the axial view of CT [10]. Other studies have shown that upper airway length (UAL) holds considerable predictive value regarding the severity of OSA [11,12]. This finding lends support to the theory that UAL could potentially contribute to the underlying mechanisms of airway collapse.

Many studies have been conducted to evaluate the factors that influence or predict CPAP compliance; however, studies on their association with the craniofacial phenotype are rare. The purpose of this study was to analyze whether craniofacial phenotype affects adherence to PAP treatment in patients diagnosed with OSA and which cephalometric parameters affect PAP adherence.

## 2. Materials and Methods

This study retrospectively reviewed the medical records of 185 patients diagnosed with OSA who were treated with a PAP device at Hallym University Dongtan Sacred Heart Hospital between July 2018 and July 2020. Patients with severe lung diseases, such as chronic obstructive pulmonary disease, upper airway obstruction due to nasal polyps, a history of oropharyngeal surgery, or those who did not undergo paranasal sinus computed tomography (PNS CT), were excluded from the study. This study was approved by the Institutional Review Board of Hallym University Dongtan Sacred Heart Hospital (IRB No. 2021-09-004).

Using medical records, the patients’ sex, age, height, weight, body mass index (BMI), underlying diseases such as hypertension or diabetes, tonsil size, Friedman palatal position [13], and questionnaire scores related to sleep quality were investigated. The Pittsburgh Sleep Quality Index (PSQI) and the Korean Epworth Sleepiness Scale (KESS) were administered before starting treatment for sleep apnea. KESS is a questionnaire that translated the existing Epworth Sleepiness Scale [14] into Korean, and there is no difference in content except that “vehicles (e.g., a car, a bus, or a train)” was changed instead of “car” in the fourth question. This questionnaire has been used in Korea as a screening tool for OSA in patients complaining of daytime sleepiness [15].

To diagnose OSA, polysomnography (PSG) was performed in all patients. Total sleep time (TST); sleep efficiency; apnea–hypopnea index (AHI); respiratory disturbance index (RDI); oxygen desaturation index (ODI); limb movements; periodic limb movements in sleep (PLMS); sleep stages such as rapid eye movement (REM) or wake; arousal index (AI); REM AHI; nadir SpO_2_; and supine AHI were investigated. The usage time and 90% pressure were investigated using the positive airway pressure records.

PNS CT was performed to examine anatomical abnormalities of the upper airway. In the midsagittal view of CT, cephalometric parameters such as H-PNS (distance between the hyoid bone and posterior nasal spine), posterior nasal spine-mandibular plane (distance between posterior nasal spine and mandibular plane, PNS-MP), H-MP (distance between the hyoid bone and mandibular plane), superior posterior airway space (distance between the posterior wall of the pharynx and the soft palate parallel to the B-Go plane, sPAS), inferior posterior airway space (distance between the posterior wall of the pharynx and the base of the tongue parallel to the B-Go plane, iPAS), and SPL were measured (Figure 1). And in the axial view of computed tomography, the anterior–posterior width (AP), lateral width (LW), square area (SA), and airway cross-sectional area (AWA) of the airway were measured at the level of the anterior–inferior corner of the second cervical vertebra [10].

Patient compliance was evaluated 3–12 months after the prescription of the PAP device. PAP devices compute the percentage of days it is used, the average use on all days, and the percentage of nights it is used for longer than 4 h. Patients who used the device for ≥ 4 h/day on at least 70% of nights were classified into the adherent group, and those who did not meet the condition or gave up were classified into the nonadherent group. Differences in clinical characteristics, PSG results, PAP device use, and cephalometric parameters measured on PNS CT were compared between the two patient groups.

SPSS (Statistical Package for the Social Sciences, Version 24.0; SPSS Inc., Chicago, IL, USA) was used for the statistical analysis. A chi-squared test, Student’s *t*-test, and one-way analysis of variance (ANOVA) were used for comparison according to PAP adherence. Analysis of covariance (ANCOVA) was used to calculate the estimated mean adjusted for sex and height. Linear regression analysis was performed to estimate the beta coefficients (with 95% confidence intervals (CI)) for AHI. Logistic regression analysis was used to estimate odds ratios (ORs) with 95% CI. The *p*-value was two-tailed, and statistical significance was set at *p* < 0.05.

## 3. Results

Among the patients who participated in the study, 139 were classified as an adherent group and 49 were classified as a nonadherent group (Table 1). In the adherent group, males accounted for 92.6% of the 126 patients, while in the nonadherent group, males accounted for 79.6% (39 patients). There was a statistically significant difference in the sex distribution between the two patient groups (*p* = 0.012). In addition, the average height of the adherent group was 172.3 ± 6.6 cm and that of the nonadherent group was 169.4 ± 7.6 cm. There was a statistically significant difference in average height between the two patient groups (*p* = 0.012). Age, weight, BMI, hypertension, diabetes, PSQI, KESS, Friedman palatal position, and tonsil size were not significantly different between the two patient groups.

Comparing the PSG results of the two patient groups before PAP treatment (Table 2), the AHI in the adherent group was 42.8 ± 23.8, which was statistically significantly higher than that in the nonadherent group, 27.8 ± 21.1 (*p*-value < 0.001). In the adherent group, mild (5 ≤ AHI < 15), moderate (15 ≤ AHI ≤ 30), and severe (AHI > 30) OSA were 17 (12.5%), 28 (20.6%), and 91 (66.9%), respectively. In the nonadherent group, there were 12 (24.5%), 24 (49.0%), and 13 (26.5%) patients, respectively (Figure 2). RDI was 46.5 ± 23.2 in the adherent group, significantly higher than 32.6 ± 20.9 in the nonadherent group (*p*-value < 0.001). ODI was 34.8 ± 23.9 in the adherent group, significantly higher than 23.5 ± 21.8 in the nonadherent group (*p*-value = 0.004). AI was 44.5 ± 20.9 in the adherent group, significantly higher than 30.6 ± 18.0 in the nonadherent group (*p*-value < 0.001). REM AHI was 43.2 ± 24.0 in the adherent group, significantly higher than 33.5 ± 24.9 in the nonadherent group (*p*-value = 0.018). The nadir SpO_2_ was 77.3 ± 9.5% in the adherent group, which was significantly lower than 81.7 ± 7.5% in the nonadherent group (*p*-value = 0.004). Supine AHI was 52.9 ± 24.4 in the adherent group, significantly higher than 38.7 ± 23.5 in the nonadherent group (*p*-value = 0.001). There were no significant differences in the TST, sleep efficiency, limb movement, PLMS, REM, or wake items between the two patient groups. When the results of using the PAP device in the two patient groups were compared, the use time was 5.2 ± 1.0 h in the adherent group, which was statistically significantly higher than 2.3 ± 0.9 h in the nonadherent group (*p*-value < 0.001). In the adherent group, 90% pressure was 9.8 ± 2.2 cmH_2_O, which was significantly higher than 8.7 ± 1.6 cmH_2_O in the nonadherent group (*p*-value < 0.001). In the Auto-PAP (APAP) device, “90% pressure” is defined as the pressure at which the patient spent 90% of the time at or below.

Cephalometric parameters measured using PNS CT in the two patient groups were compared (Table 3). H-PNS was 79.2 ± 7.2 cm in the adherent group, which was longer than 75.6 ± 8.2 cm in the nonadherent group, which was statistically significant (*p*-value = 0.004). H-MP was 21.0 ± 5.6 cm in the adherent group, significantly longer than 18.3 ± 6.1 cm in the nonadherent group (*p*-value = 0.005). The PNS-MP, sPAS, iPAS, SPL, AP, LW, SA, AWA, and AWA/SA ratios were not significantly different between the two patient groups. Because there were significant differences in the sex distribution and mean height in the clinical characteristics of the two patient groups, the estimated mean and 95% confidence interval (CI) of the cephalometric parameters adjusted for sex and height were calculated using ANCOVA (Table 4). At this time, the estimated mean of the H-MP length in the adherent group was 20.4 (95% CI 18.7–22.0) cm, which was longer than that of the nonadherent group, 18.2 (16.3–20.2) cm, which was statistically significant (*p*-value = 0.027) (Figure 3). In the case of H-PNS, the length of the adherent group was 76.9 (74.9–78.9) cm, which was longer than that of the nonadherent group of 74.9 (72.5–77.3) cm (*p*-value = 0.074). Other cephalometric parameters were not significantly different between the two patient groups.

A linear regression analysis was performed to investigate the correlation between cephalometric parameters and AHI, which is a reference point for OSA severity (Table 5). AHI was positively correlated with H-PNS, PNS-MP, H-MP, and AP and negatively correlated with LW, which was statistically significant (*p* < 0.05) (Figure 4).

Logistic regression analysis was used to obtain the OR for adherence and severe OSA according to the tertiles of the cephalometric parameters (Table 6). In the case of H-PNS, the OR of adherence in the second tertile compared to the first tertile was 2.38 (1.07–5.28). In the third tertile, the OR of adherence was 2.58 (1.14–5.81), and the OR of severe OSA was 4.86 (2.24–10.54). In the case of H-MP, the OR of adherence was 2.93 (1.25–6.86) and the OR of severe OSA was 4.00 (1.87–8.56) in the third tertile compared to the first tertile.

## 4. Discussion

The compliance rate of positive airway pressure device treatment varies depending on the study, and Korean studies have shown compliance rates of 30–40% [16,17,18]. The total compliance rate of the entire patient group (185 patients) was 73%, and the adherence group accounted for a larger portion than the nonadherent group. Patients in their 40s and 50s accounted for the highest proportion in both the adherent and nonadherent groups. Sleep apnea affects at least 10% of the middle-aged population and is associated with metabolic diseases, traffic accidents, reduced productivity at work, and higher health insurance premiums [19,20,21].

The reasons for the increase in compliance in this study are presumed to be sufficient explanations for the use of CPAP devices, the active cooperation of patients with treatment, and the development of CPAP devices. After 2018, positive airway pressure treatment was covered by national health insurance, and it was expected that the cost and access to treatment would increase, resulting in increased compliance.

In this study, when the demographic characteristics of the adherent and nonadherent groups for PAP were compared, the ratio of males in the adherent group was higher, as was height. There were no significant differences in age, BMI, underlying diseases, or subjective symptoms. Although sex has been mentioned as a factor influencing adherence to PAP, several studies have not shown consistent results [2,4]. In the case of height, it could not be ruled out that the result was due to the difference in the male to female ratio in each patient group. Therefore, as mentioned in the Results section, cephalometric parameter values were compared after adjusting for age and height using ANCOVA. Nevertheless, the nonadherent group had fewer patients than the adherent group, and there was a difference in gender distribution, which could have affected the results and is considered a limitation of this study.

Several studies have shown that the higher the value of indicators related to OSA severity, such as the AHI and ODI, the higher the PAP adherence [22,23,24]. In this study, the AHI, RDI, ODI, and AI values were higher in the adherent group, and the nadir SpO_2_ was lower, suggesting that OSA was more severe, which is consistent with the results of previous studies. However, there was no significant difference between the patient groups in subjective sleep-related symptoms measured using the PSQI and KESS before treatment, which is also consistent with the results of previous studies [2,23]. However, some studies have reported that changes in sleep symptoms before and after treatment are related to adherence [2]. However, in this study, there were no data on sleep symptoms after treatment, which is considered a limitation.

The risk of sleep apnea increases if the upper airway is structurally narrow or if the craniofacial shape is abnormal. The etiologies of sleep apnea include craniofacial morphology, soft tissue hypertrophy, sleeping posture, age, sex (male), nasal obstruction, and pharyngeal fat cells [25,26]. In patients with sleep apnea, craniofacial morphological factors can be considered through cephalometry. Cephalometry can be easily used in an outpatient setting and provides information such as measurements of the base of the skull, location of the hyoid bone, shape of the mandible, posterior airway space, shape of the tongue, and thickness of the palate [9,27].

Cephalometric parameters are generally measured using X-rays; however, in this study, they were measured using PNS CT. Unlike radiography, CT has the advantage of being closer to the actual sleeping position because it is performed while lying down. In addition, cephalometric parameters can be measured not only in the sagittal view but also in the axial view. However, in general, since the test is performed in a nonsleeping state, there is a limit to reproducing the sleeping environment. Because there are differences in respiratory drive, muscle tone, and sensitivity between sleep and wakefulness, further studies on sleeping subjects are needed.

In this study, H-PNS was longer in the patient group with high adherence to PAP, which was also found to be associated with OSA severity. H-PNS, which is the length of the upper airway [28], is correlated with sleep apnea and explains the resistance to airflow and the soft tissue of the upper glottis [29]. Previous studies have also reported that the longer the upper airway, the higher the severity of OSA [11,12,28]. As an explanation for the mechanism of this phenomenon, when the volumes of the upper airways are equal, the cross-sectional area decreases as the length of the upper airway increases. Consequently, the velocity of the airflow increases, and the negative pressure acting on the outer wall of the airway increases [12].

The H-MP represents the downward transition of the hyoid bone by the vertical length connecting the line from the convex chin point to the corner chin point and the uppermost part of the hyoid bone. The inferior hyoid bone positively correlates with airway collapsibility [8] and OSA severity [26]. The hyoid bone is connected to the muscles of the tongue and is an important structure that determines its position. In patients with OSA, it is possible that the hyoid bone is gradually displaced downward due to depression by the large tongue mass or chronically pulling the head forward to adapt to increased airway resistance from childhood [26]. In this study, we confirmed that the inferior position of the hyoid bone was significantly associated with adherence in a model corrected for sex and height and, furthermore, was associated with the severity of OSA. It is assumed that the above result was obtained because the inferior position of the hyoid bone is associated with severe OSA, and patients with severe OSA show good PAP adherence. However, further research is required to elucidate the exact mechanism.

Several existing studies have demonstrated that SPL is increased [28,30] and sPAS and iPAS are decreased [28,31] in patients with OSA. However, in this study, SPL, sPAS, and iPAS did not show significant differences according to PAP adherence and OSA severity. However, when measuring SPL using CT, an error may occur because the tip of the uvula is not located in the midsagittal view. In addition, since the length of the posterior airway space may change due to the collapsing force generated by airflow during breathing [8], caution is needed in interpreting the results of this study. This point is considered a limitation in the measurement of the above parameters using CT.

Shigeta et al. reported that decreased retroglossal airway area ratio (AWA/SA) in the axial CT view could be useful for evaluating patients with OSA [10]. Therefore, in this study, we investigated whether the retroglossal airway index was related to adherence; however, there was no significant difference between the two patient groups. As the upper airway becomes more collapsible when the mouth is open during sleep, we can assume that the axial parameters found in our study would become less pronounced during awakening [32]. However, AP showed a positive correlation with AHI, and LW showed a negative correlation, suggesting that additional research on the relationship between axial parameters and OSA severity is required.

Several studies have been conducted on the cephalometric parameters of OSA patients according to ethnicity. In Caucasians, the anterior cranial base length was decreased, the superior posterior airway space was reduced, and the soft palate length was increased. Some studies in Asians showed a decrease in the degree of maxillary or mandibular protrusion. However, inferiorly displaced hyoid tended to increase in patients with OSA regardless of ethnicity [33]. Also, in a study comparing the association between airway collapsibility and cephalometric parameters in Japanese-Brazilians and whites, MP-H and upper airway length showed significant correlations in both ethnic groups [34]. The patients who participated in this study were all Koreans (Asian), and it seems consistent with the results of previous studies in that MP-H and H-PNS showed a positive correlation with OSA severity.

## 5. Conclusions

This study establishes a significant correlation between an inferiorly placed hyoid bone in adult patients with obstructive sleep apnea (OSA) and their positive airway pressure (PAP) adherence and disease severity. Our observations reveal that PAP adherence group in OSA patients exhibit inferiorly placed hyoid bones and longer upper airway lengths, which are also closely linked to severe OSA. Consequently, the assessment of hyoid position (H-MP) and upper airway length (H-PNS) would serve as reliable predictors for PAP adherence in clinical practice. However, further prospective studies are required to confirm this hypothesis.

## Figures and Tables

**Figure 1 jpm-13-01196-f001:**
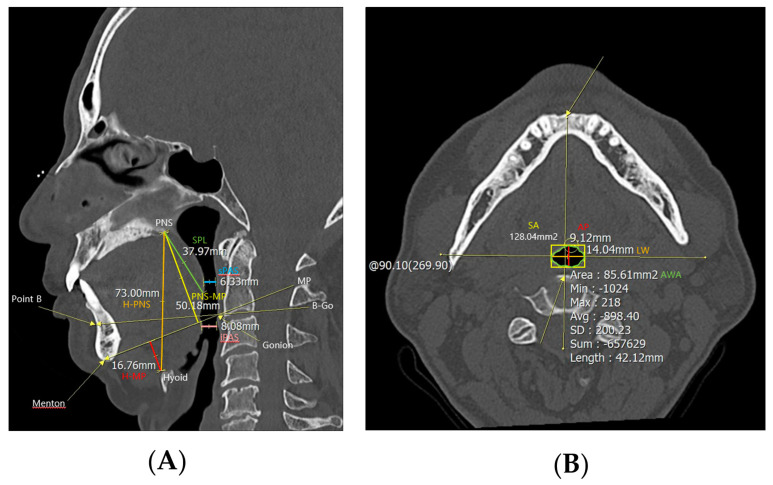
Cephalometric parameters measured using PNS CT. H−PNS, PNS−MP, H−MP, sPAS, iPAS, and SPL were measured in (**A**) the midsagittal view of the PNS CT, and AP, LW, SA, and AWA were measured in (**B**) the axial view. Abbreviations: H−PNS, hyoid–posterior nasal spine; P–MP, posterior nasal spine–mandibular plane; H–MP, hyoid–mandibular plane; sPAS, superior posterior airway space; iPAS, inferior posterior airway space; SPL, soft palate length; AP, anterior–posterior width; LW, lateral width; SA, square area; AWA, airway cross–sectional area.

**Figure 2 jpm-13-01196-f002:**
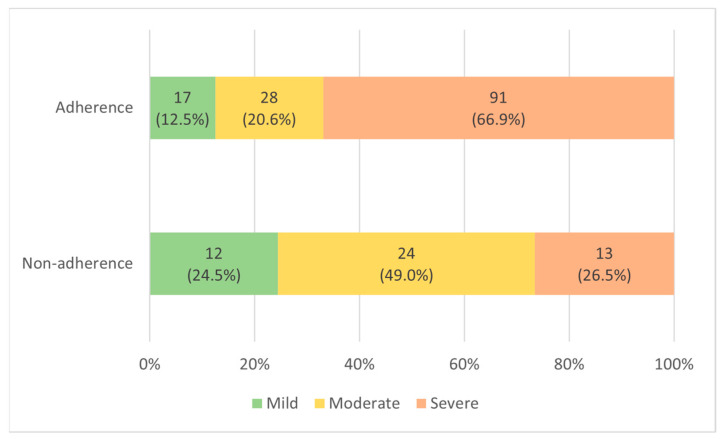
Distribution according to OSA severity in adherent and nonadherent group.

**Figure 3 jpm-13-01196-f003:**
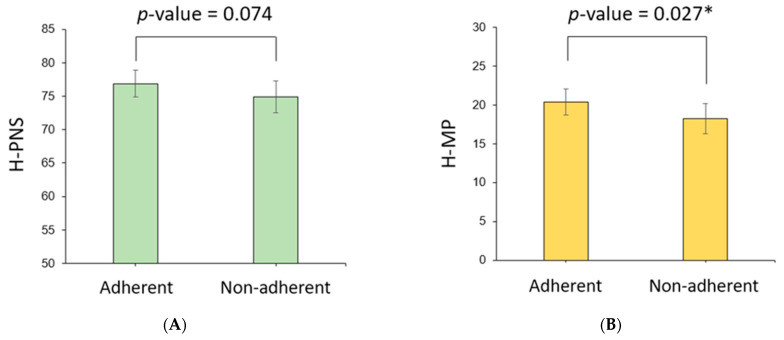
Comparison of cephalometric parameters between adherent and nonadherent groups. In the model adjusted for height and performance using ANCOVA, the adherent group (**A**) tended to have longer H-PNS and (**B**), and H-MP were significantly longer than the nonadherent group. * *p* < 0.05.

**Figure 4 jpm-13-01196-f004:**
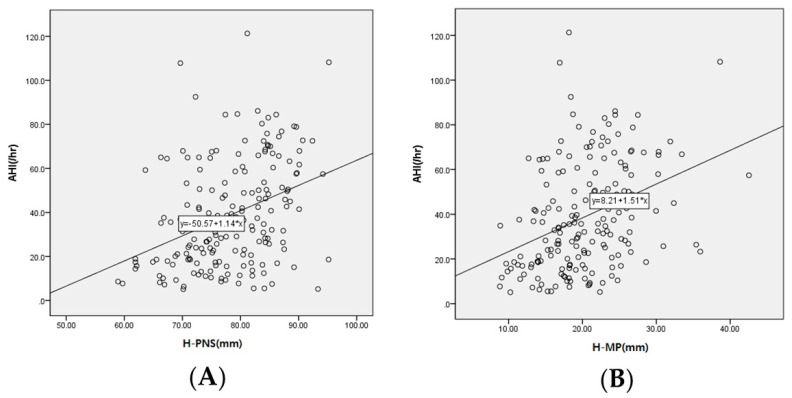
Scatterplot between cephalometric parameters and AHI. The *y*-axis of the scatterplot represents the AHI, and the *x*-axis represents (**A**) H-PNS and (**B**) H-MP.

**Table 1 jpm-13-01196-t001:** Demographic and anatomical findings of adherent and nonadherent group.

		Adherent (N = 136)	Nonadherent (N = 49)	*p*-Value
Gender (male/female)		126 (92.6)/10 (7.4)	39 (79.6)/10 (20.4)	0.012
Age (years)		49.4 ± 9.6	49.7 ± 11.4	0.852
Height (cm)		172.3 ± 6.6	169.4 ± 7.6	0.012
Weight (kg)		81.6 ± 13.2	79.1 ± 14.5	0.265
BMI (kg/m^2^)		27.4 ± 3.5	27.4 ± 3.8	0.947
HTN		60 (44.1)	20 (41.7)	0.768
DM		17 (12.5)	5 (10.4)	0.702
PSQI		8.4 ± 3.1	8.6 ± 3.0	0.777
KESS		9.6 ± 5.0	8.8 ± 3.8	0.275
Friedman palatal position	1	3 (2.2)	1 (2.0)	0.302
	2	9 (6.6)	4 (8.2)	
	3	86 (63.2)	35 (71.4)	
	4	38 (27.9)	9 (18.4)	
Tonsil size	1	120 (88.2)	41 (83.7)	0.214
	2	14 (10.3)	6 (12.2)	
	3	2 (1.5)	1 (2.0)	
	4	0 (0.0)	1 (2.0)	

Values are presented as mean ± standard deviation or count (percentage). Abbreviations: BMI, body mass index; DM, diabetes mellitus; HTN, hypertension; PSQI, Pittsburgh Sleep Quality Index; KESS, Korean Epworth Sleepiness Scale.

**Table 2 jpm-13-01196-t002:** PSG findings and PAP records of adherent and nonadherent group.

	Adherent	Nonadherent	*p*-Value
TST (minutes)	338.5 ± 35.6	349.1 ± 32.9	0.072
Sleep efficiency (%)	86.8 ± 11.2	85.3 ± 10.4	0.433
AHI (/hour)	42.8 ± 23.8	27.8 ± 21.1	<0.001
RDI (/hour)	46.5 ± 23.2	32.6 ± 20.9	<0.001
ODI (/hour)	34.8 ± 23.9	23.5 ± 21.8	0.004
Limb movement (/hour)	17.1 ± 18.9	16.7 ± 16.4	0.915
PLMS (/hour)	9.0 ± 17.5	8.2 ± 15.0	0.762
REM (%)	12.7 ± 4.8	12.6 ± 4.5	0.860
Wake (%)	12.0 ± 11.0	13.2 ± 10.4	0.536
AI (/hour)	44.5 ± 20.9	30.6 ± 18.0	<0.001
REM AHI (/hour)	43.2 ± 24.0	33.5 ± 24.9	0.018
Supine AHI (/hour)	52.9 ± 24.4	38.7 ± 23.5	0.001
Nadir SpO_2_ (%)	77.3 ± 9.5	81.7 ± 7.5	0.004
Usage time (hours)	5.2 ± 1.0	2.3 ± 0.9	<0.001
* 90% pressure (cmH_2_O)	9.8 ± 2.2	8.7 ± 1.6	<0.001

Values are presented as mean ± standard deviation. Abbreviations: PSG, polysomnography; PAP, positive airway pressure; TST, total sleep time; AHI, apnea–hypopnea index; RDI, respiratory disturbance index; ODI, oxygen desaturation index; PLMS, periodic limb movements in sleep; REM, rapid eye movement; AI, arousal index. * In Auto-PAP (APAP) device, “90% pressure” is defined as the pressure at which the patient spent 90% of the time at or below.

**Table 3 jpm-13-01196-t003:** Cephalometric parameters of adherent and nonadherent group.

	Adherent	Nonadherent	*p*-Value
H-PNS (mm)	79.2 ± 7.2	75.6 ± 8.2	0.004
PNS-MP (mm)	50.8 ± 5.5	50.4 ± 6.0	0.697
H-MP (mm)	21.0 ± 5.6	18.3 ± 6.1	0.005
sPAS (mm)	9.1 ± 2.5	9.6 ± 2.7	0.300
iPAS (mm)	11.2 ± 3.2	11.4 ± 3.5	0.704
SPL (mm) †	39.9 ± 5.0	38.8 ± 4.0	0.189
AP (mm) †	15.5 ± 4.3	15.4 ± 4.4	0.846
LW (mm) †	24.6 ± 7.1	25.5 ± 6.3	0.459
SA (mm^2^) †	382.3 ± 150.8	395.0 ± 155.8	0.638
AWA (mm^2^) †	209.3 ± 94.1	222.2 ± 94.2	0.440
AWA/SA †	0.55 ± 0.10	0.56 ± 0.06	0.446

Values are presented as mean ± standard deviation. † After excluding missing values, there were 116 patients in the adherent group and 44 patients in the nonadherent group. Abbreviations: H-PNS, hyoid–posterior nasal spine; P-MP, posterior nasal spine–mandibular plane; H-MP, hyoid–mandibular plane; sPAS, superior posterior airway space; iPAS, inferior posterior airway space; SPL, soft palate length; AP, anterior–posterior width; LW, lateral width; SA, square area; AWA, airway cross-sectional area.

**Table 4 jpm-13-01196-t004:** Comparative model of cephalometric parameters adjusted using ANCOVA.

	Adherent	Nonadherent	*p*-Value
H-PNS (mm)	76.9 (7.9–78.9)	74.9 (72.5–77.3)	0.074
PNS-MP (mm)	49.3 (47.7–50.9)	49.7 (47.8–50.9)	0.638
H-MP (mm)	20.4 (18.7–22.0)	18.2 (16.3–20.2)	0.027
sPAS (mm)	9.7 (8.9–10.4)	10.0 (9.1–10.9)	0.395
iPAS (mm)	10.9 (9.9–11.9)	11.4 (10.2–12.5)	0.406
SPL (mm) †	38.3 (36.8–39.8)	37.8 (36.0–39.5)	0.540
AP (mm) †	15.0 (13.6–16.5)	15.2 (13.5–16.8)	0.879
LW (mm) †	24.7 (22.4–27.0)	25.9 (23.3–28.5)	0.331
SA (mm^2^) †	367.5 (317.7–417.2)	392.7 (334.7–450.7)	0.357
AWA (mm^2^) †	198.0 (167.3–228.8)	219.2 (183.4–255.1)	0.211
AWA/SA †	0.54 (0.51–0.57)	0.55 (0.52–0.59)	0.329

This model was adjusted by gender and height. Values are presented as estimated mean (95% confidence interval, CI). † After excluding missing values, there were 116 patients in the adherent group and 44 patients in the nonadherent group. Abbreviations: ANCOVA, analysis of covariance.

**Table 5 jpm-13-01196-t005:** Linear regression analysis for the correlation between cephalometric parameters and AHI.

	B	SE	*p*-Value
H-PNS (mm)	1.14	0.22	<0.001
PNS-MP (mm)	0.64	0.32	0.045
H-MP (mm)	1.51	0.28	<0.001
sPAS (mm)	−0.15	0.70	0.834
iPAS (mm)	0.04	0.54	0.937
SPL (mm)	0.01	0.01	0.189
AP (mm)	1.07	0.45	0.018
LW (mm)	−0.57	0.28	0.044
SA (mm^2^)	0.00	0.01	0.962
AWA (mm^2^)	−0.02	0.02	0.361

Abbreviations: B, unstandardized regression coefficient; SE, standard error.

**Table 6 jpm-13-01196-t006:** Logistic regression analysis for the correlation between tertile of cephalometric parameters and adherence or severe OSA.

		Adherence		AHI > 30	
		OR (95% CI)	*p*-Value	OR (95% CI)	*p*-Value
H-PNS (mm)	<74.8	1 (reference)		1 (reference)	
	74.8–83.0	2.38 (1.07–5.28)	0.033	1.92 (0.94–3.93)	0.073
	>83.0	2.58 (1.14–5.81)	0.022	4.86 (2.24–10.54)	<0.001
PNS-MP (mm)	<48.5	1 (reference)		1 (reference)	
	48.5–52.7	0.74 (0.33–1.63)	0.454	0.85 (0.42–1.72)	0.651
	>52.7	0.96 (0.42–2.18)	0.918	1.42 (0.69–2.94)	0.341
H-MP (mm)	<17.5	1 (reference)		1 (reference)	
	17.5–22.8	1.62 (0.75–3.48)	0.219	1.68 (0.82–3.41)	0.154
	>22.8	2.93 (1.25–6.86)	0.013	4.00 (1.87–8.56)	<0.001
sPAS (mm)	<8.1	1 (reference)		1 (reference)	
	8.1–10.3	1.97 (0.79–4.87)	0.144	1.31 (0.64–2.68)	0.452
	>10.3	0.52 (0.24–1.13)	0.099	1.01 (0.50–2.04)	0.988
iPAS (mm)	<9.8	1 (reference)		1 (reference)	
	9.8–12.8	0.77 (0.34–1.75)	0.534	0.72 (0.35–1.47)	0.365
	>12.8	0.70 (0.31–1.57)	0.382	0.74 (0.36–1.53)	0.420
SPL (mm)	<37.4	1 (reference)		1 (reference)	
	37.4–41.1	0.92 (0.41–2.06)	0.837	1.25 (0.59–2.67)	0.562
	<41.1	2.25 (0.90–5.63)	0.084	1.41 (0.66–3.05)	0.375
AP (mm)	<13.0	1 (reference)		1 (reference)	
	13.0–16.6	1.71 (0.73–4.02)	0.221	0.96 (0.44–2.06)	0.910
	>16.6	1.39 (0.61–3.20)	0.435	0.79 (0.37–1.71)	0.558
LW (mm)	<21.5	1 (reference)		1 (reference)	
	21.5–27.5	1.12 (0.47–2.67)	0.789	0.49 (0.22–1.06)	0.071
	>27.5	0.87 (0.38–2.00)	0.738	0.51 (0.23–1.10)	0.086
SA (mm^2^)	<303.5	1 (reference)		1 (reference)	
	303.5-433.0	1.81 (0.73–4.46)	0.197	0.83 (0.39–1.79)	0.635
	>433.0	0.82 (0.36–1.85)	0.631	0.83 (0.39–1.79)	0.635
AWA (mm^2^)	<164.2	1 (reference)		1 (reference)	
	164.2–237.0	1.00 (0.42–2.36)	>0.999	0.46 (0.21–1.01)	0.053
	>237.0	0.79 (0.34–1.84)	0.580	0.58 (0.27–1.28)	0.179

## Data Availability

The datasets generated and analyzed during the current study are not publicly available due to institutional restrictions, but are available from the corresponding author upon reasonable request.

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
