# Peer review of "Craniofacial Phenotype in Obstructive Sleep Apnea and Its Impact on Positive Airway Pressure (PAP) Adherence"

_jpm, 2023, doi:10.3390/jpm13081196_

Round 1

Reviewer 1 Report

The topic of the study (Craniofacial Phenotype in Obstructive Sleep Apnea and its Impact on Positive Airway Pressure Adherence) is interesting. The association between anthropometric indices and the severity of OSA have been investigated previously, but their impact on PAP therapy adherence is relatively novel. In spite of this strength, it seems that some revisions are needed in the article.

  Firstly, what is meant by medication in line 112 and 90% pressure in the text and table? Second, the method of evaluating the adherence should be clearly explained in the article. Third, it is recommended to assess the effect of OSA severity on PAP compliance by categorizing each group based on AHI. Furthermore, ethnic differences in anthropometric characteristics and specific features of study population should be included in the discussion. Finally, the small sample size especially in nonadherent group and gender distribution of the study are limitations that make it difficult to distribute the results. These should be mentioned as limitations. 

Minor editing of English language required.

Reviewer 2 Report

Dear authors,  The article addresses an interesting topic, but:

1. To define more clearly what is compliance and adherence to treatment of the patient with apnea - in the introduction chapter

2. Row 84 - to describe what Korean ESS is...more precisely, what are the differences compared to classic ESS

3. Row 112 - Device is more correct than medication

4. Row 219 –I suggest you -  associated with metabolic diseases

5. In the discussions Chapter  the details and comparative studies regarding cephalometric parameters are important

 6. During the conclusions, I would have liked it to be clearer about the particularities of the craniofacial phenotype in the patient with SASO AND THE CORRELATION  WITH  ADHERENCE TO CPAP THERAPY...that there should be correspondence between the title, aim and conclusions

....what is now in the conclusions, are general data, known!!!

Round 2

Reviewer 2 Report

congratulations for your work!!!